# Influence of Multiplex PCR in the Management of Antibiotic Treatment in Patients with Bacteremia

**DOI:** 10.3390/antibiotics12061038

**Published:** 2023-06-10

**Authors:** Alina-Ioana Andrei, Daniela Tălăpan, Alexandru Rafila, Gabriel Adrian Popescu

**Affiliations:** 1Faculty of Medicine, “Carol Davila” University of Medicine and Pharmacy, 050474 Bucharest, Romania; 2“Prof. Dr. Matei Balș” National Institute of Infectious Diseases, 021105 Bucharest, Romania

**Keywords:** blood cultures, multiplex PCR testing system, antimicrobial therapy

## Abstract

The multiplex PCR assay can be a helpful diagnostic tool for patients with bacteremia. Herein, we assessed the impact of a Blood Culture Identification Panel (BCID) on both the diagnosis and treatment of patients with bacteremia. We performed a retrospective study using laboratory and clinical data to evaluate the impact of syndromic testing using a multiplex PCR testing system (BioFire^®^ FilmArray) for the management of patients with bloodstream infections. BCID detected the pathogen in 102 (87.9%) samples out of the 116 positive blood cultures tested. The average time from the blood culture collection to the communication of the molecular test result was 23.93 h (range: 10.67–69.27 h). The main pathogen detected was *Klebsiella pneumoniae* (17.6%). The antimicrobial therapy was changed in accordance with the BCID results in 28 (40.6%) out of the 69 cases, wherein the treatment could have been theoretically adjusted. This allowed the adjustment of the therapy to be performed 1305.1 h faster than it would have been possible if conventional diagnostic methods had been used; this was the case for only 35.1% of the time gained if treatment was adjusted for all patients with positive BCID. Thus, although molecular tests can make a difference in the management of bloodstream infections, there is room for improvement in the clinical application of BCID results.

## 1. Introduction

Sepsis is a life-threatening organ dysfunction caused by a dysregulated host response to infection [1]. Current diagnostic guidelines, namely, qSOFA (quick Sequential Organ Failure Assessment) and SOFA (Sequential Organ Failure Assessment) scores, have established specific criteria for selecting patients with acute infection who are at risk of developing sepsis [2]. Depending on the time, quality of care, and method of data collection, the mortality rate in sepsis varies from 17.9% to 52% [3,4,5]. Blood cultures are an important tool in the diagnosis of sepsis; existing data indicate that 30–40% of patients with sepsis have positive blood cultures [6]. Melville et al. [7] reported a 20% relative decrease in mortality when the post-Surviving Sepsis Campaign guidelines (second edition) period was compared to the pre-guideline period. Furthermore, it was observed that delayed administration of antimicrobials beyond 3 h is associated with increased mortality [5]. To decrease the mortality rate in septic patients, the Surviving Sepsis Campaign developed a group of measures that should be adopted in the first hour, called the bundle. Of these, the collection of blood cultures is the most important step [8], which should be collected prior to antimicrobial administration. 

Given the high mortality rate, both rapid identification and antimicrobial susceptibility testing are crucial. To start faster appropriate antimicrobial treatment, positive identification of the pathogen should be made in a timely manner. Using the conventional microbiological diagnosis algorithm, the time from the blood culture collection to bacterial identification and susceptibility testing results ranges between 60 and 72 h [9,10]. 

New microbiological methods of diagnosis and susceptibility testing have been developed over the years. Pathogens in blood cultures can now be identified sooner either directly from a positive blood culture or from colonies after subculturing on agar plates using matrix-assisted laser desorption ionization-time-of-flight (MALDI-TOF). The entire process takes 5–45 min, depending on the quality of the culture and the need to carry out more operations for the processing of the specimen tested [11]. MALDI-TOF can accurately identify the majority of the common pathogens, including the most frequent Gram-positive (*Staphylococcus* spp., *Streptococcus* spp., and *Enterococcus* spp.) and Gram-negative (*Escherichia coli*, *Klebsiella* spp., *Acinetobacter* spp., *Pseudomonas aeruginosa*, *Morganella* spp., and *Salmonella* spp.) bacteria. However, this method has some limitations, such as the inability to differentiate *Shigella* spp. from *E. coli* [12,13]. This technology can also be used to accelerate susceptibility testing. Verroken et al. [14] observed a reduction of approximately 27% of the time needed to obtain the susceptibility testing results when compared to the classical method. Another new diagnostic tool is the multiplex polymerase chain reaction test FilmArray—the BioFire Blood Culture Identification Panel (BCID). It can detect 27 targets in about 1 h (Table 1). This method has been proven to be very reliable, with results comparable to those of conventional blood culture identification, but faster. Altun O. et al. [10] showed in a recent study that the FilmArray BCID panel covered all the microorganisms in 91.6% of the positive blood cultures analyzed. Patient care data published from a tertiary hospital in Belgium showed a decrease of about 10 h in the time from growth to optimal antimicrobial therapy (OAT) when using the BCID panel (4 h and 39 min) compared to the standard procedure (14 h and 41 min). The standard algorithm consisted of positive blood cultures, strain identification, and susceptibility testing using MALDI-TOF [15]. Messacar et al. [16] revealed in their study that the median OAT decreased from 60.2 h to 26.7 h when the BCID panel was used. 

In addition to the identification of microorganisms, the BCID panel can detect three resistance genes (Table 1). This feature is even more important when treating patients admitted to the intensive care unit because of the high resistance profile of the microorganisms involved in these infections. However, because there are some bacterial/yeast species not identified by this test—such as *Stenotrophomonas maltophilia*, *Klebsiella aerogenes*, *Salmonella* spp., *Enterococcus faecium*/*faecalis* and *Candida auris*—a new version was developed. The BCID2 panel can now detect 43 targets, including 33 pathogens (26 bacterial species and 7 fungal species) and 10 resistance markers (Table 1). According to a recent study, the BCID2 panel shows good concordance with other methods (MALDI-TOF) in monomicrobial cultures (94.0% correctly identified organisms). Furthermore, when compared to its previous version, it has a better ability to differentiate between the two enterococcal species [17]. Peri et al. [18] analyzed the BCID2 panel and concluded that if it was implemented in their daily workflow, the results would have been available 9.69 h sooner than with using MALDI-TOF (95% CI: 7.85–11.53) for antimicrobial identification, and the resistance genes would have been detected 27.8 h earlier than with using VITEK 2 system (95% CI: 23.05–32.55).

Current antimicrobial stewardship guidelines recommend the use of rapid diagnostic tests to quickly obtain a correct diagnosis to administer appropriate antimicrobial therapy, thereby improving patient outcomes [19]. The effect of rapid molecular tests on the outcome of patients with bacteremia was analyzed in a systematic review and meta-analysis. Timbrook et al. [20] concluded that the mortality rate was significantly lower when these tests were used than the classic microbiological diagnosis (OR: 0.66; 95% CI: 0.54–0.80). Moreover, it was the lowest when the results were used together with an antimicrobial stewardship program (OR: 0.64; 95% CI: 0.51–0.79). Current studies show that bacterial co-infections and secondary infections are less frequent in patients with COVID-19 than in patients with influenza [21]. Secondary bacterial infections seem to be more frequent with a longer length of hospitalization, both in COVID-19 and influenza. Patients with seasonal influenza usually develop secondary infections after six days of symptoms [22]. Pasquini et al. highlighted in their study that the incidence of secondary infections in COVID-19 patients emerged from the third day of admission and continued to increase over the duration of the admission [23]. 

In patients with COVID-19, bacterial infections are more often superinfections in patients already admitted to the hospital, especially in the intensive care unit [24,25]. A different profile of bloodstream bacteria was reported to be isolated from patients with SARS-CoV-2 infection compared to other patients, with a greater dominance of hospital-acquired bacteria. Bayo et al. identified a significant difference in hospital-acquired bacteria in bloodstream infections (30.5% in non-COVID-19 patients and 95.5% in COVID-19 patients) (*p* < 0.001) [26]. In our clinic, BCID, and shortly thereafter, BCID 2 became available during the COVID-19 pandemic. The aim of this study was to determine the influence of the BCID panel results on the clinical management of patients admitted to our hospital during the COVID-19 pandemic.

## 2. Results

### 2.1. Sample Characteristics

During the analysis period, 116 positive blood cultures were tested using the BCID panel. Of these, 35 (30.2%) samples were collected from patients admitted to the intensive care unit (ICU), and 81 (69.8%) samples were collected from the infectious disease wards. Of the total blood cultures tested, 14 (12.1%) samples were tested negative by the Film Array, and in 102 (87.9%) blood cultures, the pathogen was identified using this molecular test, which correlated 100% with the culture result. 

### 2.2. Analysis of Samples Tested Negative by the BCID Panel 

Fourteen samples tested as “not detectable”. In six (42.9%) cases, pathogenic bacteria were isolated using conventional culturing methods. Four of them could not be identified by either BCID or BCID2 because these pathogens were not targeted by the BCID panel, and two (two strains of Stenotrophomonas maltophilia) were not identified because, at the time of isolation, these microorganisms were not included in the BCID panels. In the remaining eight (57.1%) cases, some contaminant agents were isolated. These microorganisms were not detected because they were not in the BCID panel. The results are shown in Figure 1.

### 2.3. Profile of the Pathogens Detected

The main pathogens isolated were Klebsiella pneumoniae (19 samples, 17.6%), followed by Escherichia coli (17 samples, 15.7%), Acinetobacter baumannii (14 samples, 13%), and Enterococcus spp. (12 samples, 11.1%). Nine blood cultures tested positive for Staphylococcus aureus. Candida parapsilosis and Candida albicans were isolated from three patients each. Two patients showed evidence of Candida auris bacteremia. Seven (6.9%) of the bacteria identified using the genetic assay were considered contaminated. (Figure 1)

### 2.4. Antimicrobial Susceptibility

Sixteen strains of Klebsiella pneumoniae were resistant to ceftazidime and seven were resistant to colistin. All Acinetobacter baumannii strains were susceptible to colistin. Three Escherichia coli strains were resistant to ceftazidime, but all of them maintained their susceptibility to both carbapenems and aminoglycosides. Three strains of Staphylococcus aureus were methicillin-resistant, whereas none of them were resistant to glycopeptides or linezolid. None of the enterococcal species detected were vancomycin-resistant. Table 2 and Table 3 summarize the isolated strains and their susceptibility and resistance profiles.

There was a 100% correlation between the detection of resistance genes using BCID/BCID2 and the results of antimicrobial susceptibility testing. The mecA gene was detected in three out of nine patients with Staphylococcus aureus bacteremia using the BCID panel. As shown in Table 2, three strains of Staphylococcus aureus were found to be resistant to oxacillin based on antimicrobial susceptibility testing. BioFire identified five K. pneumoniae strains with both CTX-M and NDM genes and seven KPC-positive isolates. These detected resistance genes were correlated with resistance to meropenem, as detected by antimicrobial susceptibility testing (Table 3). One strain tested positive for both CTX-M and OXA-48 resistance genes and one tested positive for only the OXA-48 gene. All types of carbapenemases and ESBL (Extended Spectrum Beta-Lactamase) were identified using phenotypic methods. None of the Enterococcus spp. tested positive for vanA/vanB genes.

### 2.5. Influence of the BCID Panel on Antimicrobial Treatment

The influence of the BCID panel on the administered antimicrobial treatment was analyzed in 89 cases (87.3%). In six cases, this information was not available because either the patients died before the blood cultures were positive and/or the BCID panel result was obtained, or the patients were transferred to other hospitals. In 28 cases (31.5%), the antimicrobial therapy was adjusted immediately after the BCID results were communicated by the microbiology department, whereas in 18 cases (20.2%), clinicians preferred to wait for the susceptibility testing results before making a decision. The antimicrobial therapy was maintained after the BCID results were available in 20 cases (22.5%), and in 12 (13.5%) of them, the therapy remained unchanged because it was already adequate for the detected pathogen. The antimicrobial treatment could have been adjusted theoretically in 69 of the 89 cases. Our analysis showed that the therapy was changed according to the BCID results in 28 of the 69 cases (40.6%). The results of the analysis are presented in Table 4 and Table 5 and Figure 2.

### 2.6. The Influence of BCID on Reducing the Time to Obtaining the Blood Culture Results

The average time from the blood culture collection to communicating the BCID test result to the clinician was 23.93 h (range: 10.67–69.27 h), whereas the average time from blood culture collection to the communication of the susceptibility testing results was 80.43 h (range: 36.32–231.77 h). By using rapid molecular tests, the time to establish the etiological diagnosis and decide on the correct antimicrobial susceptibility profile was 56.5 h sooner compared to classic methods.

The use of BCID results for treatment adjustment was performed in 28 of 69 patients, which accounted for 1305.1 h in which the treatment was adapted, 35.1% of what could have been in the case of the maximum use of the BCID results.

## 3. Discussion

In our study, the average time from collection to results communication was 23.93 h, which was longer than the time identified in recent studies by Graff et al. [27] (19 h, 95% CI: 17–21) and Berinson et al. [17] (13.6 h), but lower than the mean time to the BCID results obtained by Sparks et al. [28] (24.6 h, N=51) or Messacar et al. [16] (26.7 h). Nevertheless, the BCID panel results were significantly quicker than those of the classical microbiological identification algorithm. The BCID panel results could influence the clinician’s antimicrobial therapy decisions in approximately one-quarter of the cases, which is consistent with recently published data [29]. The number of cases in which the BCID panel could not identify the microorganisms was small, and because in most of these cases, contaminant agents were cultured, the impact on patients’ outcomes was minor.

The detection of carbapenemase-producing bacteria is an important tool and deciding factor when establishing the appropriate antimicrobial therapy, even more in the setting of 100% concordance with phenotypic testing. In our study, BCID detected both OXA-48- and KPC-producing *Klebsiella pneumoniae,* and the usage of ceftazidime–avibactam was validated. Five *Klebisella pneumoniae* strains tested positive for NDM; in these cases, ceftazidime–avibactam was no longer efficient, and in the absence of cefiderocol, either colistin, tigecycline, or both can be used [30,31].

Our study has some limitations. A significant number of blood cultures in our study were collected from patients admitted to the ICU. Given the high circulation of Gram-negative bacilli in our ICU, we investigated the impact of BCID/BCID2 on Gram-negative infections, rather than infections induced by Gram-positive cocci. Another limitation of this study is connected to the current practice within our hospital, which is the avoidance of using BCID data in a significant ratio. This aspect made it difficult to highlight the real impact of this test in the management of patients with invasive infections. 

In conclusion, the BCID panel is a great asset for clinicians because it can guide them in tailoring the appropriate antimicrobial therapy. In our case, there is still room for improvement, which is going to be a task for the Antimicrobial Stewardship Team. The team members should obtain a daily list of the positive blood culture molecular test results from the microbiology department and then personally discuss these results with clinicians to explain the significance of the results and identify appropriate treatment regimens. A follow-up visit should be performed to ensure that the changes have been made in accordance with the Antimicrobial Stewardship Team recommendations and also to clinically and biologically reevaluate the patient.

## 4. Materials and Methods

This study was conducted between March 2020 and November 2022 at the National Institute of Infectious Diseases “Prof. Dr. Matei Bals”, Bucharest, Romania, which is a tertiary care hospital.

We performed a retrospective study using laboratory and clinical data to evaluate the influence of syndromic testing using a multiplex PCR testing system (BioFire^®^ FilmArray^®^, BioFire Diagnostics, Salt Lake City, UT, USA) in the management of patients with bloodstream infections (BSI). Laboratory data were extracted from the microbiology laboratory database, and the clinical data were extracted from the hospital database. The institutional review board granted access to the data without the need for individual informed consent because the data were analyzed anonymously. This study was conducted in accordance with the ethical standards of the 1964 Declaration of Helsinki and its amendments. The following data were collected for each patient: demographic characteristics, hospital ward at the time of BSI, date and time of blood culture collection, the date and time of communication of the positive result, detected pathogens and resistance genes (if that was the case) by the BioFire System, the time of first communication of the positive result of blood culture identification with BCID, the final result with susceptibility testing, and the time of antibiotic change (if any).

### Laboratory Methods

Blood cultures were performed as requested by a physician for patients suspected of BSI (fever 38 °C or higher). In this study, consecutive cases of positive blood cultures, one per patient (between 1 March 2020 and 30 November 2022) were included. Blood culture bottles (FA Plus, FN Plus, SA, SN, and PF plus; BioMérieux S.A., Marcy l’Etoile, France) were incubated at 37 °C, and those identified as positive by the BacT/Alert 3D (BioMérieux Inc., Durham, NC, USA) were removed and processed for 24 h, 7 days/week.

Positive culture bottles were processed according to the standard protocol of the microbiology laboratory (Gram-staining examination, subculture on solid media, identification, and antimicrobial susceptibility testing). The BioFire^®^ FilmArray^®^ Blood Culture Identification (BCID) Panel or the BCID2 Panel (BCID/BCID2, BioMérieux, BioFire Diagnostics, Salt Lake City, UT, USA) was also used for molecular syndromic testing of positive blood cultures. One molecular syndromic test was performed per patient. The BCID was used from the beginning of the study until the end of August 2021 (for 11 positive blood cultures), when the new panel type (BCID2) became available in our laboratory (there were 105 positive samples with BCID2). BCID/BCID2 testing was performed as per the manufacturer’s guidelines. Briefly, the following protocol was carried out: after a positive blood culture bottle was flagged positive, Gram staining was performed, followed by culturing using Columbia agar with sheep blood (ThermoFisher Scientific™-Oxoid, Wesel, Germany) and chocolate agar plates (BioMérieux S.A., Marcy l’Etoile, France). If yeasts were detected by Gram staining, they were cultured on Sabouraud agar plates (ThermoFisher Scientific™-Oxoid, Wesel, Germany). After interpreting the Gram staining, the BioFire BCID Panel was run for all rods (Gram-positive and Gram-negative), yeasts, and Gram-positive cocci in diplo and/or in chains. The results of the pathogen identification (and, if applicable, also the resistance gene) became available in approximately 1 h and 15 min and were communicated immediately by phone to the physician on duty and were updated in the patient’s electronic medical records. 

The information about the isolation and precise identification of the microorganism was available after incubation for 4–8 h at 35–37 °C using the MALDI Biotyper^®^ (Bruker Daltonics GmbH & KG, Bremen, Germany). The antimicrobial and antifungal susceptibility testing followed immediately after identification, and it was performed using the Vitek^®^ 2Compact (BioMérieux S.A., Marcy l’Etoile, France) or Micronaut system (MERLIN Diagnostika GmbH, Bornheim, Germany). The results were communicated the next day after appropriate interpretation according to the European Committee on Antimicrobial Susceptibility Testing (EUCAST) guidelines [32]. 

Disk diffusion susceptibility and *E*-tests were used where required, according to the local protocol. ESBL and/or carbapenemase phenotypes, as identified by Vitek 2 Compact or Micronaut susceptibility testing, were confirmed using the ESBL combination disk test (Rosco Diagnostica A/S, Taastrup, Denmark) combined with AmpC detection (Rosco Diagnostica A/S, Taastrup, Denmark), if necessary. The combination disk test with meropenem alone and in combination with various inhibitors (Rosco Diagnostica A/S, Taastrup, Denmark) was performed according to the EUCAST guidelines for the detection of resistance mechanisms (version 2.0, July 2017) [33]. 

The time taken to communicate the results was calculated from the time of specimen collection to the time when the BCID/BCID2 result became available in the patients’ medical records. The time taken to provide effective antimicrobial therapy was calculated from the time of specimen collection to the first dose of the antimicrobial agent that tested as susceptible, according to the EUCAST criteria. The time taken to provide optimal therapy was calculated from the time of specimen collection to the time when the first dose of the optimal antimicrobial therapy was administered. Optimal antimicrobial therapy was defined based on the predefined antimicrobial guidelines developed in conjunction with the Antimicrobial Stewardship Team for each BCID/BCID2 target. The antimicrobial treatment was considered adequate if it was active against the specific strain but had the narrowest spectrum, the lowest risk of developing *Clostridioides difficile* infection, and if it was in accordance with local guidelines. In addition to these recommendations, drug allergies, possible organ dysfunctions (hepatic and renal), drug interactions, and the source of the bacteremia were taken into account when evaluating whether the treatment was optimal or not. For example, if methicillin-susceptible *Staphylococcus aureus* was detected then the antimicrobial treatment selected would have been oxacillin, not vancomycin or ceftriaxone. Furthermore, if *Enteroccus faecalis* was detected, ampicillin would be the appropriate treatment. If a KPC- or OXA-48-producing bacteria was detected, ceftazidime–avibactam was considered a good choice, and if an NDM-producing strain was identified, colistin or tigecycline or both would be appropriate choices. Although both cefiderocol and ceftazidime–avibactam plus aztreonam are alternative treatment options for infections caused by NDM-positive isolates, they are not considered treatment options because neither cefiderocol nor intravenous aztreonam is available in our country.

A blood culture result was considered a contamination if the microorganism detected was a well-known contaminant agent, such as coagulase-negative staphylococci, *Corynebacterium* spp., *Bacillus* spp. other than *Bacillus anthracis*, and *Cutibacterium acnes*, and if it was isolated from a single blood culture bottle. These bacterial species were considered true positives if they were detected in at least two blood culture sets and from patients with immunosuppression or implanted medical devices. The extracted clinical and laboratory data were entered into a database and analyzed using Epi Info version 7.2. The data related to the patients’ medication and the time when they were administered were extracted from the hospital’s electronic medical records.

## 5. Conclusions

Molecular tests offer true support for the management of patients with severe infections. Our study showed that there is room for improvement in the use of BCID results in the clinic, and antimicrobial stewardship measures will lead to an increase in the clinician’s confidence in the new rapid molecular tests, which will help to provide the most appropriate treatment to patients.

## Figures and Tables

**Figure 1 antibiotics-12-01038-f001:**
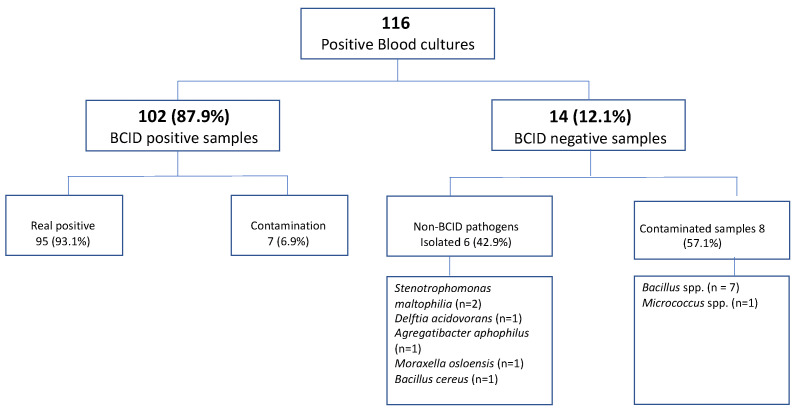
Sample characteristics.

**Figure 2 antibiotics-12-01038-f002:**
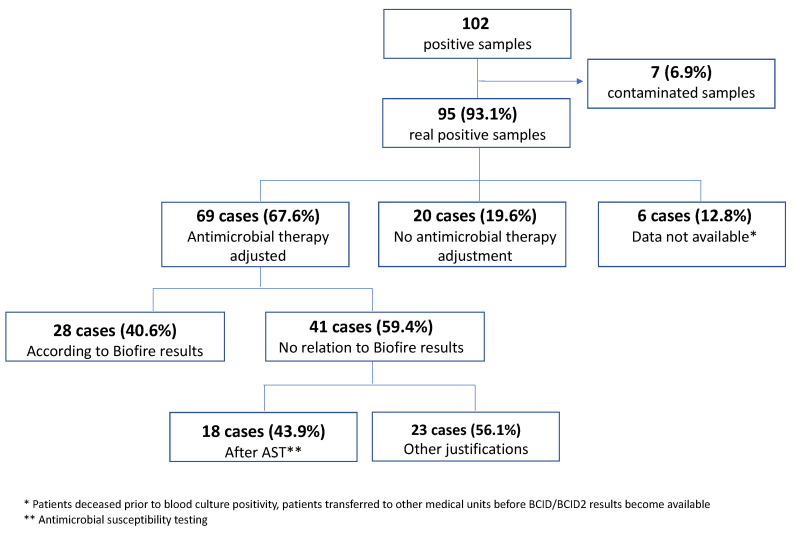
BCID as a driving factor for antimicrobial treatment changes (40.6% of all patients with treatment changes).

**Table 1 antibiotics-12-01038-t001:** BioFire Blood Culture Identification Panel.

Category	BioFire Blood Culture Identification Panel	BioFire Blood Culture Identification Panel 2
	Target	Target
Gram-positive bacteria	*Staphylococcus* spp. *Staphylococcus aureus* *Streptococcus* spp. *Streptococcus agalactiae* *Streptococcus pyogenes* *Streptococcus pneumoniae* *Enterococcus* spp. *Listeria monocytogenes*	*Staphylococcus* spp. *Staphylococcus aureus* ***Staphylococcus epidermidis*** ***Staphylococcus lugdunensis*** *Streptococcus* spp. *Streptococcus agalactiae* *Streptococcus pyogenes* *Streptococcus pneumoniae* *Enterococcus* spp. ***Enterococcus faecalis*** ***Enterococcus faecium*** *Listeria monocytogenes*
Gram-negative bacteria	Enterobacterales *Escherichia coli* *Enterobacter cloacae complex* *Klebsiella oxytoca* *Klebsiella pneumoniae* *Serratia marcescens* *Proteus* spp. *Haemophilus influenzae* *Acinetobacter baumannii* *Pseudomonas aeruginosa* *Neisseria meningitidis*	Enterobacterales *Escherichia coli* *Enterobacter cloacae complex* *Klebsiella oxytoca* *Klebsiella pneumoniae group* ***Klebsiella aerogenes*** *Serratia marcescens* *Proteus* spp. ***Salmonella* spp.** *Haemophilus influenzae* *Acinetobacter baumannii* *Pseudomonas aeruginosa* *Neisseria meningitidis* ***Stenotrophomonas maltophilia*** ***Bacteroides fragilis***
Yeast	*Candida albicans* *Candida glabrata* *Candida parapsilosis* *Candida tropicalis* *Candida krusei*	*Candida albicans* *Candida glabrata* *Candida parapsilosis* *Candida tropicalis* *Candida krusei* ** *Candida auris* ** ***Cryptococcus* (*C.**neoformans*/*C.**gattii*)**
Resistance genes	*mecA* *vanA*/*vanB* KPC	*mecA/C* *mecA/C* and **MREJ** (MRSA) *vanA/vanB* KPC **IMP** **NDM** **OXA-48-like** **VIM** *mcr-1* **CTX-M**

Note: Words in bold indicate new targets added in Panel 2 compared to the first one.

**Table 2 antibiotics-12-01038-t002:** Antimicrobial susceptibility/resistance profile of the main Gram-positive pathogens identified.

Antimicrobial	*Staphylococcus aureus* N		*Enterococcus faecalis* N		*Enterococcus faecium* N	
	Susceptible	Resistant	Susceptible	Resistant	Susceptible	Resistant
Oxacillin	6	3	-	-	-	-
Ampicillin	-	-	11	0	0	1
Gentamicin *	9	0	10	1	0	1
Erythromycin	4	5	-	-	-	-
Clindamycin	5	4	-	-	-	-
Linezolid	9	0	11	0	1	0
Vancomycin	9	0	11	0	1	0
Trimethoprim-sulfamethoxazole	9	0	-	-	-	-

* Gentamicin high resistance for *Enterococcus* spp.

**Table 3 antibiotics-12-01038-t003:** Antimicrobial susceptibility/resistance profile of the main Gram-negative pathogens identified.

Antimicrobial	*E. coli* N		*K. pneumoniae* N		*A. baumannii* N		*P. aeruginosa* N	
	Susceptible	Resistant	Susceptible	Resistant	Susceptible	Resistant	Susceptible	Resistant
Ampicillin	3	14	-	-	-	-	-	-
Piperacillin-tazobactam	16	1	3	16	-	-	1 I	2
Ceftazidime	14	3	3	16	-	-	2 I	2
Meropenem	17	0	7 (5 S + 2 I)	12	0	14	2	2
Amikacin	17	0	4	15	3	11	3	1
Ciprofloxacin	12	5	4	15	0	14	2 I	2
Colistin	17	0	12	7	14	0	4	0
Trimethoprim-sulfamethoxazole	7	10	7	12	1 I	13	-	-

S = Susceptible; I = susceptible at increased exposure.

**Table 4 antibiotics-12-01038-t004:** Influence of the BCID Blood Culture Identification Panel impact on antimicrobial treatment.

Category	Number (%)
Therapy adjusted after BCID panel results	28 (27.5)
Therapy changed without any relation to BCID panel results	23 (22.5)
Therapy changed after susceptibility testing results	18 (17.6)
No antimicrobial therapy adjustment	20 (19.6)
Contamination	7 (6.9)
No data available	6 (5.9)
Total	102 (100)

**Table 5 antibiotics-12-01038-t005:** Time saved by BCID for adequate antimicrobial treatment.

Category	Number	Time to BCID Results (h)	Time to Classic Methods Results (h)	Time Saved (h)
Therapy adjusted after BCID results	28/116 (24.1%)	20.77 ± 10.7 (SD = 10.7)	67.38 (SD = 22.5)	1305.1
Therapy not adjusted after BCID results, but could have been adjusted	41/116 (35.3%)	26.14 (SD = 14.05)	85.06 (SD = 41.06)	2415.7
Therapy not adjustable after BCID results	33/116 (28.5%)			NA
Negative BCID	14/116 (12.1%)			NA
Total/mean	116 (100%)	23.9	80.4	3720.8

SD = standard deviation; NA = not applicable.

## Data Availability

Not applicable.

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
