# Peer review of "Influence of Multiplex PCR in the Management of Antibiotic Treatment in Patients with Bacteremia"

_antibiotics, 2023, doi:10.3390/antibiotics12061038_

Round 1

Reviewer 1 Report

 Impact of multiplex PCR in the management of antibiotic treatment in patients with bacteremia. The study aimed to determine the impact of the BCID Panel results on the 125 clinical management patients admitted to the hospital during the SARS-CoV-2 pandemic. The authors analyzed 116 positive blood cultures and tested the commercial BCID panel. They claim that BCID panel results are significantly quicker when compared to the classical microbiological identification protocol. Also, the conclusion is that the BCID panel isan asset for clinicians because it can apply appropriate antimicrobial therapy. 

The manuscript lacks deep analysis, such as an association of the antimicrobial susceptibility testing of the isolated strains with BCID panel results. 

There is no data on the clinical characteristics of the patients, age, sex, or clinical indication. Were the samples collected in consecutive cases, months, years, or days? 

It was unclear if the BCID test was performed directly in positive blood cultures or patients’ blood samples.

The authors did not show a clear correlation of data in Tables 2 and 3 (that summarize the susceptibility and resistance profiles of the isolated strains) with the results of the BCID panel.

Tables 4 and 5 and Figure 2 are very confusing ways to present the results. 

The BCID time-saved for adequate antimicrobial treatment in hours (1305.1) is misleading and refers to 28 out of 69 patients.

The discussion section is very unsatisfied and shallow.

Minor comments:

-Line 89: correction: faecalis.

-Line 297: “was used from the beginning of the study until the end of August 2021, when the new panel type (BCID2) became available in our laboratory.”-

How many patients’ samples were submitted to the BCID and the new panel type BCID2. Please inform if panel 2 was used for all patients since panel 2 displays more pathogens and resistant genes. Since the BCIDcan detect more 7-gram positive bacterial targets, 5-gram negative targets, 3 yeast species, and 7 resistance genes, BCID negative samples may be due to the absence of bacterial targets.

-Line 136: “Out of the total blood cultures tested, 14 (12.1%) samples were not detectable by the Film Array, and in 102 (87.9%) blood cultures the pathogen was identified using this test.

The not detected microorganisms data is because they were not a target in the BCID panel? Non BCID isolates, what are they? 

-Line 221: “The results of the analysis are presented in Table 4, Table 5 and Figure 2”. The authors should explain the data in the text or combine data from Tables 4 and 5. 

- Line: 351  Our study showed that there is room for improvement in the use of BCID results in the clinic.” The authors should provide some suggestions for this comment.

English editing is required

Author Response

Dear Reviewer,

Thank you for your valuable feedback. We addressed all the points that you suggested, and the responses are in the attached document.

Kind regards,

Daniela Talapan

Reviewer 2 Report

Thank you for asking me to review this manuscript. Within the manuscript there is interesting data on experience of use of multiplex PCR and its clinical impact. However, the manuscript includes a lot of other information which could be summarised and referenced. The introduction amounts to an entire review of sepsis. This could be drastically edited, with retention of references on key points. Other sections are more balanced. 

Minor changes are required

Author Response

Dear Reviewer,

Thank you for your comments on our paper.

We revised and edited whole paper, including the introduction, keeping in mind your suggestions.

Kind regards,

Daniela Talapan

Round 2

Reviewer 1 Report

The manuscript Influence of multiplex PCR in the management of antibiotics treatment in patients with bacteremia in this second version has improved the description and results that flow with tables and figures. I consider the scope of the information to be clear and may interest readers of Antibiotics.

Minor comments

Line 85: Table 1

Table 3: indicate: N- Number of susceptibility or resistant isolates

Author Response

Dear reviewer, 

Thank you very much for your comments which helped us improve our paper. Also, thank you for appreciation of our effort and work.

Sincerely yours, 

Daniela Talapan

Reviewer 2 Report

Thank you for making revisions to the manuscript. I still think that the Introduction is more wide-ranging than it needs to be and could be shortened with the reader looking up references. 

Minor changes needed only

Author Response

(The authors gave the same response as above.)

Round 3

Reviewer 2 Report

Thank you for your further attempt to edit the introduction. 

Author Response

Dear Reviwer, 

Thank you for your kind words and for tremendous help during the editing this paper. 

Sincerely yours, 

The authors
